# Differentiable Discrete Elastic Rods for Real-Time Modeling of Deformable Linear Objects

**Yizhou Chen**    **Yiting Zhang**    **Zachary Brei**    **Tiancheng Zhang**
**Yuzhen Chen**    **Julie Wu**    **Ram Vasudevan**

Department of Robotics, University of Michigan, Ann Arbor, MI 48109, United States

{yizhouch, yitzhang, breizach, zhangtc, yuzhench, jwuxx, ramv}@umich.edu

https://roahmlab.github.io/DEFORM/

**Abstract:** This paper addresses the task of modeling Deformable Linear Objects (DLOs), such as ropes and cables, during dynamic motion over long time horizons. This task presents significant challenges due to the complex dynamics of DLOs. To address these challenges, this paper proposes differentiable Discrete Elastic Rods For deformable linear Objects with Real-time Modeling (DEFORM), a novel framework that combines a differentiable physics-based model with a learning framework to model DLOs accurately in real-time. The performance of DEFORM is evaluated in an experimental setup involving two industrial robots and a variety of sensors. A comprehensive series of experiments demonstrate the efficacy of DEFORM in terms of accuracy, computational speed, and generalizability when compared to state-of-the-art alternatives. To further demonstrate the utility of DEFORM, this paper integrates it into a perception pipeline and illustrates its superior performance when compared to the state-of-the-art methods while tracking a DLO even in the presence of occlusions. Finally, this paper illustrates the superior performance of DEFORM when compared to state-of-the-art methods when it is applied to perform autonomous planning and control of DLOs.

**Keywords:** Deformable Linear Objects Modeling, Physics-Informed Learning, Differentiable Simulation

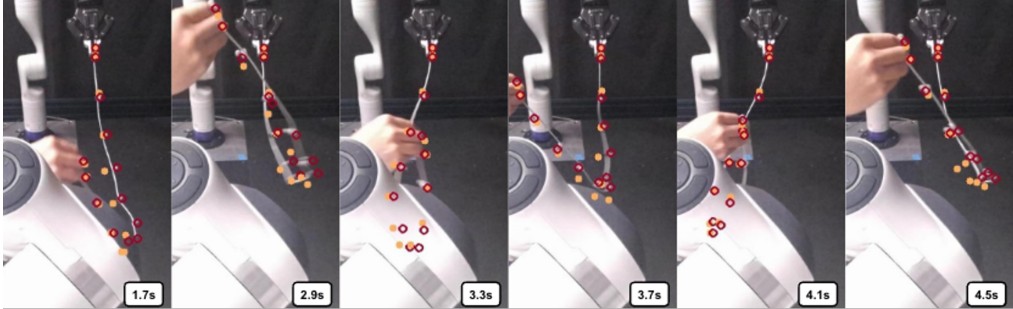

Figure 1: This paper introduces DEFORM, a novel framework that combines a differentiable physics-based model with a learning framework to model and predict dynamic DLO behavior accurately in real-time. The figure shows DEFORM's predicted states (yellow) and the actual states (red) for a DLO over 4.5 seconds at 100 Hz. Note that the prediction is performed recursively, without requiring access to ground truth or perception during the process. A video of related experiments can be found in the supplementary material.

## 1   Introduction

Robotic manipulation tasks such as surgical suturing or vehicle wire harness assembly [1, 2, 3] are challenging problems because they require accurately and dynamically manipulating DLOs over

8th Conference on Robot Learning (CoRL 2024), Munich, Germany.

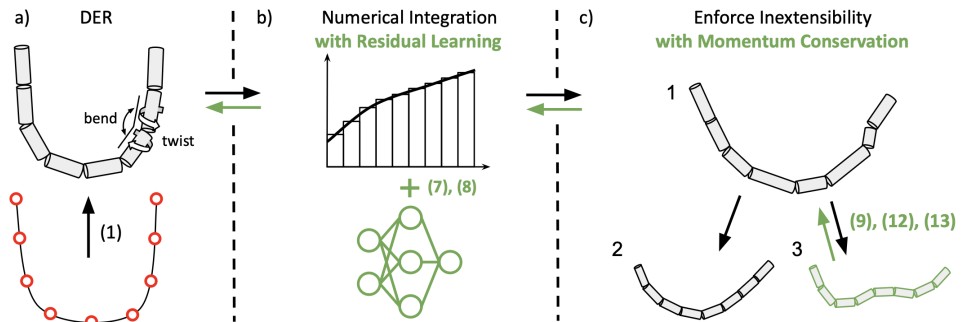

Figure 2: Overview of DEFORM contributions (green). a) DER models discretize DLOs into vertices, segment them into elastic rods, and model their dynamic propagation. DEFORM reformulates DER into Differentiable DER (DDER) which describes how to compute gradients from the prediction loss, enabling efficient system identification and incorporation into deep learning pipelines. b) To compensate for the error from DER's numerical integration, DEFORM introduces residual learning via DNNs. c) $1 \rightarrow 2$: DER enforces inextensibility, but this does not satisfy classical conservation principles. $1 \rightarrow 3$: DEFORM enforces inextensibility with momentum conservation, which allows dynamic modeling while maintaining simulation stability.

long time horizons ($> 1s$) [4]. Nonlinear dynamics make manipulating DLOs challenging because computationally expensive models are required for accurate prediction. Further, practical applications such as vehicle wire harness assembly often result in DLO occlusions, meaning perception systems cannot robustly estimate the entire DLO configuration. For robotic manipulators to successfully manipulate DLOs, a novel modeling method capable of quick and accurate predictions of dynamic DLO behavior, even in the presence of occlusions, is needed.

To model DLOs, physics-based approaches can be used, but they suffer a tradeoff between costly computation and low accuracy when predicting dynamic movements. State-of-the-art learning-based modeling approaches have used deep neural networks (DNNs) to model interaction propagation in DLOs [5, 6, 7]. Despite their success, these methods require large datasets of real-world manipulation data, which can be impractical to collect and are not robust to variations in DLO properties. Alternatively, using a physics-based model as a prior for DNNs is promising due to increased sampling efficiency and generalizability [8, 9, 10, 11, 12]. However, to the best of our knowledge, this approach has not been applied to predict behavior of DLOs during dynamic motions.

A recent promising physics-based modeling method is Discrete Elastic Rods (DER). This approach *qualitatively* and efficiently describes the 3-D behavior of DLOs [13, 14]. Despite its potential for physics-informed learning, applying DER theory to model DLOs in the real-world is challenging for three reasons. First, tuning DER model parameters to match a real-world DLO is difficult. Second, computing analytical gradients for DER models is challenging. Third, numerical errors arise from discrete numerical integration of DER models and from enforcing DLO inextensibility. These drawbacks affect the accuracy of the DER model, limiting the utility of embedding a DER model in a learning-based framework. In particular, a lack of analytical gradients prevent learning over multiple steps, which reduces the accuracy of long horizon predictions.

To address the above challenges, this paper introduces **D**ifferentiable Discrete **E**lastic Rods **F**or Deformable Linear **O**bjects with **R**eal-time **M**odeling (**DEFORM**), a novel framework that uses a differentiable physics-based learning model, with real-time inference speed, to accurately predict dynamic DLO behavior over long time horizons. As illustrated in Figure 2, the main contributions of DEFORM are four-fold. First, a novel DLO model is presented, called the Differentiable Discrete Elastic Rod (DDER) model that makes DER models differentiable with respect to any variables. Second, a numerical integration error compensation method that uses DNNs is developed to improve accuracy when numerically integrating DER models. Third, a novel inextensibility enforcement algorithm is presented for DER models that guarantees momentum conservation, thereby enabling accurate prediction of dynamic DLO behavior. Fourth, a comprehensive set of experiments are presented, which illustrate that DEFORM has superior performance in terms of accuracy, computational speed, and sampling efficiency. We also demonstrate the utility of DEFORM for robust DLO tracking and precise shape-matching manipulation tasks.

## 2  Related Work

**Learning-Based Modeling.** Latent dynamics modeling approaches, such as graph neural networks (GNN) [15] and Bidirectional LSTM (Bi-LSTM) [6], reason over vertices spatially to formulate latent dynamics of DLOs for predicting their behavior in 2-D. Unfortunately, as we illustrate in this paper through real-world experiments, these methods are unable to accurate describe the behavior of DLOs during dynamic motions in 3D.

**Physics-Based Modeling.** Traditionally, DLOs have been modeled using physics-based approaches. Examples include mass-spring system models [16, 17, 18], Position-Based Dynamics (PBD) models [19, 20, 21], Discrete Elastic Rod (DER) models [13, 14, 22, 23], or Finite Element Method (FEM) models [24, 25, 26]. Although easy to implement, the mass-spring system imposes unnecessary stiffness to enforce inextensibility, resulting in instability during simulation. FEM models accurately modeling DLO behavior, but are impractical for real-time prediction due to their prohibitive computational costs. PBD and DER methods have been applied in planning and control frameworks for manipulation tasks due to their computational efficiency [20, 21, 27, 28, 29]. However, PBD models are sensitive to parameter selection and assume quasi-static behavior. As illustrated in this paper, this results in low accuracy during dynamic manipulation. Though DER models have been qualitatively validated to describe the 3D behavior of DLOs, we illustrate that their quantitative performance while describing dynamic motion is lacking.

**Differentiable Modeling Framework.** Differentiable models are computational frameworks where every operation is differentiable with respect to the input parameters. Importantly, a differentiable model enables integration into a learning framework to improve learning and can enable efficient parameter auto-tuning via gradient-based optimization during training. There are differentiable models for deformable objects, such as DiffCloth [16, 17] and XPBD [21], but these models tend to suffer from numerical instability. This makes it challenging to apply these modeling frameworks while trying to perform parameter identification for real-world DLO manipulation. To address these limitations, this paper proposes a differentiable DER model, called DDER, which is incorporated into a learning framework to achieve numerical stability. This results in better predictive performance for dynamic DLO behavior when compared to state-of-the-art methods and enables auto-tuning of DLO parameters using limited real-world data.

## 3  Preliminaries

**Deformable Linear Object (DLO) Model.** To model a DLO, we separate the DLO into an indexed set of $n$ vertices at each time instance. The ground truth locations of vertex $i$ at time $t$ is denoted by $\mathbf{x}_t^i \in \mathbb{R}^3$ and the set of all $n$ vertices is denoted by $\mathbf{X}_t$. Let $\boldsymbol{M}^i \in \mathbb{R}^{3 \times 3}$ denote the mass matrix of vertex $i$. Our objective is to model the dynamic behavior of a DLO that is being controlled by a robot with following assumption: the first and last edge of the DLO are each held by a distinct end effector. Note this assumption is commonly made during DLO manipulation [30, 20, 19, 7, 31]. Using this assumption, let the orientation and position of the two end effectors at time $t$ be denoted by $\mathbf{u}_t \in \mathbb{R}^{12}$, which we refer to as the input. The velocity of the vertices of the DLO can be denoted as $\mathbf{V}_t = (\mathbf{X}_t - \mathbf{X}_{t\text{-}1})/\Delta t$. Note that $\mathbf{V}_t$ is an approximation of the actual velocity. Finally, note that we distinguish between ground truth elements and predicted elements by using the circumflex symbol (*e.g.*, $\mathbf{X}_t$ is the ground truth set of vertices at time $t$ and $\hat{\mathbf{X}}_t$ is the predicted set of vertices at time $t$).

**Modeling DLOs with DER for Dual Manipulation.** DLOs primarily exhibit three distinct deformation modes: bending, twisting, and stretching [13]. This paper focuses on DLOs that are inextensible (*e.g.*, cables and wires), and therefore do not experience stretching. DER models introduce a single scalar $\theta_t^i \in \mathbb{R}$ to represent bending and twisting in $\mathbb{R}^3$. This parameter describes the potential energy $P(\mathbf{X}_t, \mathbf{u}_t, \boldsymbol{\theta}_t, \boldsymbol{\alpha})$ of the DLO that arises due to bending or twisting, where $\boldsymbol{\alpha}$ denotes the vector of the material properties of each vertex of the DLO (*i.e.*, their mass, bending modulus, and twisting modulus) [13, Section 4.2]. Notably, DER models assume that the DLO reaches an equilibrium state between each simulation time step to obtain $\boldsymbol{\theta}_t^*$:

$$\boldsymbol{\theta}_t^*(\mathbf{X}_t, \boldsymbol{\alpha}) = \arg\min_{\boldsymbol{\theta}_t} P(\mathbf{X}_t, \mathbf{u}_t, \boldsymbol{\theta}_t, \boldsymbol{\alpha}) \tag{1}$$

Once the potential energy expression is derived, then the restorative force experienced during deformation is set equal to the negative of the gradient of the potential energy expression with respect to the vertices. Therefore, the equations of motion of the DLO are:

$$M\ddot{\mathbf{X}}_t = -\frac{\partial P(\mathbf{X}_t, \mathbf{u}_t, \boldsymbol{\theta}_t^*(\mathbf{X}_t, \boldsymbol{\alpha}), \boldsymbol{\alpha})}{\partial \mathbf{X}_t} \tag{2}$$

After solving (1), one can numerically integrate this formula to predict the velocity and position by applying the Semi-Implicit Euler method:

$$\hat{\mathbf{V}}_{t+1} = \hat{\mathbf{V}}_t - \Delta_t M^{-1} \frac{\partial P(\hat{\mathbf{X}}_t, \boldsymbol{\theta}_t^*(\hat{\mathbf{X}}_t, \boldsymbol{\alpha}), \boldsymbol{\alpha})}{\partial \mathbf{X}_t}, \tag{3}$$

$$\hat{\mathbf{X}}_{t+1} = \hat{\mathbf{X}}_t + \Delta_t \hat{\mathbf{V}}_{t+1}. \tag{4}$$

(3)-(4) can be recursively solved to simulate the behavior of the DER model of a DLO.

**Inextensibility Enforcement.** During simulation, the DLO's length may change due to numerical errors, but this is undesirable when modeling inextensible DLOs. To ensure constant length, DER methods project a computed solution onto a constraint set that enforces inextensibility[32]. However, this projection operation can introduce instabilities during simulation of dynamic behavior because it does not ensure that momentum is conserved. This paper adopts a constraint solver based on a PBD model [33], which is described in Section 4.3, to enforce inextensiblity while conserving momentum. This improves prediction accuracy compared to classical methods.

## 4  Methodology

This section first introduces DDER, a reformulation of DER to ensure differentiability, which enables efficient DER model parameter estimation using real-world data. Next, this section describes how to incorporate a learning-assisted integration method to improve the speed of DLO simulation and accuracy. Finally, this section describes a position-based constraint that enforces inextensibility while preserving momentum, which results in

---

**Algorithm 1** One Step Prediction with DEFORM

**Inputs**: $\hat{\mathbf{X}}_t, \hat{\mathbf{V}}_t, \mathbf{u}_t$
1: $\boldsymbol{\theta}_t^*(\mathbf{X}_t) = \arg\min_{\boldsymbol{\theta}_t} P(\mathbf{X}_t, \boldsymbol{\theta}_t, \boldsymbol{\alpha})$   $\triangleright$ Section.4.1
2: $\hat{\mathbf{V}}_{t+1} = \hat{\mathbf{V}}_t - \Delta_t M^{-1} \frac{\partial P(\mathbf{X}_t, \boldsymbol{\theta}_t^*(\mathbf{X}_t, \boldsymbol{\alpha}), \boldsymbol{\alpha})}{\partial \mathbf{X}_t}$   $\triangleright$ (3)
3: $\hat{\mathbf{X}}_{t+1} = \hat{\mathbf{X}}_t + \Delta_t \hat{\mathbf{V}}_{t+1} + \text{DNN}$   $\triangleright$ Section.4.2
4: Inextensibility Enforcement on $\hat{\mathbf{X}}_{t+1}$   $\triangleright$ Section.4.3
5: $\hat{\mathbf{V}}_{t+1} = (\hat{\mathbf{X}}_{t+1} - \hat{\mathbf{X}}_t)/\Delta_t$   $\triangleright$ Velocity Update
**return** $\hat{\mathbf{X}}_{t+1}, \hat{\mathbf{V}}_{t+1}$

---

improved numerical stability during simulation. These results are combined in a prediction algorithm that is described in Algorithm 1. This algorithm can be run over multiple steps to generate an accurate prediction of the state of the DLO over a long time horizon.

### 4.1  Differentiable Discrete Elastic Rods (DDER)

We first develop a reformulation of DER such that the DER model is differentiable with respect to any model variables (i.e., material properties, and DNN weights). We call this extension Differentiable Discrete Elastic Rods (DDER). To do this, we implement DDER in PyTorch [34] by taking advantage of its automatic differentiation functionality. Unfortunately, simulating the DDER model to predict $\hat{\mathbf{X}}_{t+1}$ using (3) and (4) also requires solving an optimization problem (1). As a result, to minimize the prediction loss, we need to solve a bi-level optimization problem at each time $t$. For instance, if we train a learning model to identify the material parameters by minimizing a prediction loss between experimentally collected ground truth data $\mathbf{X}_{t+1}$ and our model prediction $\hat{\mathbf{X}}_{t+1}$ for each time $t$, then the following bi-level optimization problem needs to be solved:

$$\min_{\boldsymbol{\alpha}} \quad \|\mathbf{X}_{t+1} - \hat{\mathbf{X}}_{t+1}(\boldsymbol{\theta}_t^*)\|_2 \tag{5}$$

$$\boldsymbol{\theta}_t^* = \arg\min_{\boldsymbol{\theta}_t} P(\hat{\mathbf{X}}_t, \mathbf{u}_t, \boldsymbol{\theta}_t, \boldsymbol{\alpha}) \tag{6}$$

Note that $\hat{\mathbf{X}}_{t+1}$ depends on the previously computed $\boldsymbol{\theta}_t^*$ due to (3) and (4). To compute the gradient of this optimization problem with respect to $\boldsymbol{\alpha}$, one can perform implicit differentiation, which we implement using Theseus [35].

## 4.2 Integration Method with Residual Learning

To compensate for numerical errors that arise from discrete integration methods and to improve the numerical stability of the algorithm that we use to simulate the DDER model, we incorporate a learning-based framework into the integration method as follows:

$$\hat{\mathbf{V}}_{t+1} = \hat{\mathbf{V}}_t - \Delta_t M^{-1}\Big(\frac{\partial P(\hat{\mathbf{X}}_t, \boldsymbol{\theta}_t^*(\hat{\mathbf{X}}_t, \boldsymbol{\alpha}), \boldsymbol{\alpha})}{\partial \mathbf{X}_t} + \mathrm{DNN}(\hat{\mathbf{X}}_t, \boldsymbol{\alpha})\Big), \tag{7}$$

$$\hat{\mathbf{X}}_{t+1} = \hat{\mathbf{X}}_t + \Delta_t\Big(\hat{\mathbf{V}}_{t+1} + \mathrm{DNN}(\hat{\mathbf{X}}_t, \hat{\mathbf{V}}_t, \boldsymbol{\alpha})\Big), \tag{8}$$

where DNN denotes a deep neural network. We utilize a structure similar to residual learning [36], with one major difference: we define a shortcut connection that is equal to the integration method that is derived from the DDER physics-based modeling. Unlike pure learning methods, the short cut connection grounds the DNN in physical laws while also leveraging data-driven insights. As shown in the ablation study results, this approach significantly improves modeling performance. Additioanl discussion of section can be found in Appendix B.5. Implementation details of the DNN structure can be found in Appendix B.1. **Note that this contribution can also be applied to standard DER models, but does not improve performance as much as it does when combined with our DDER modeling approach as we illustrate in this paper.**

## 4.3 Improved Inextensiblity Enforcement with PBD

To enforce inextensibility *and* conserve momentum after simulating the DLO forward for a single time step (8), we enforce an additional constraint on the DLO in a manner similar to the PBD method [33]. First, we define a constraint function between spatially successive vertices at each time step:

$$\mathrm{C}(\hat{\mathbf{x}}_{t+1}^i, \hat{\mathbf{x}}_{t+1}^{i+1}) = |\,||\hat{\mathbf{x}}_{t+1}^i - \hat{\mathbf{x}}_{t+1}^{i+1}||_2 - ||\bar{\mathbf{e}}_i||_2| \tag{9}$$

where $\bar{\mathbf{e}}$ denotes the edge between vertices $i$ and $i+1$ when the DLO is undeformed. Note that this constraint is non-zero if the distance between successive vertices of the DLO changes. The PBD method iteratively modifies the positions of each of the vertex using a pair of correction terms, $\Delta\hat{\mathbf{x}}_{t+1}^i$ and $\Delta\hat{\mathbf{x}}_{t+1}^{i+1}$, until the following constraint is satisfied:

$$\mathrm{C}(\hat{\mathbf{x}}_{t+1}^i + \Delta\hat{\mathbf{x}}_{t+1}^i, \hat{\mathbf{x}}_{t+1}^{i+1} + \Delta\hat{\mathbf{x}}_{t+1}^{i+1}) < \epsilon, \tag{10}$$

where $\epsilon > 0$ is a user-specified threshold. To calculate the correction terms, PBD typically applies a Taylor Approximation of the constraint and sets this equal to zero (*i.e.*, the correction to the vertex locations are chosen to ensure that the constraint is satisfied). The correction term is then given by:

$$\Delta\hat{\mathbf{x}}_{t+1}^i = C(\hat{\mathbf{x}}_{t+1}^i, \hat{\mathbf{x}}_{t+1}^{i+1})\frac{\hat{\mathbf{x}}_{t+1}^{i+1} - \hat{\mathbf{x}}_{t+1}^i}{||\hat{\mathbf{x}}_{t+1}^{i+1} - \hat{\mathbf{x}}_{t+1}^i||_2} \tag{11}$$

Notably, this correction to the vertex location does not ensure that the linear and angular momentum is preserved. To ensure that linear and angular momentum is conserved after the correction is applied, one needs to ensure that the following pair of equations are satisfied:

$$\sum_{i=3}^{n-2} \boldsymbol{M}^i \cdot \Delta\hat{\mathbf{x}}_{t+1}^i = \mathbf{0} \text{ and } \sum_{i=3}^{n-2} \mathbf{r}^i \times (\boldsymbol{M}^i \cdot \Delta\hat{\mathbf{x}}_{t+1}^i) = \mathbf{0} \tag{12}$$

where $\mathbf{r}^i$ is the vector between $\hat{\mathbf{x}}_{t+1}^i$ and a common rotation center. Note in particular that if the mass matrix associated with each vertex is not identical, then the correction proposed in (11) leads to a violation of (12). To address this issue, we modify the correction proposed in (11) by multiplying it by a $\beta \in \mathbb{R}$ that is inversely correlated with the weight ratio between successive vertices (i.e., let $\beta^i = \frac{\boldsymbol{M}^{i+1}}{(||\boldsymbol{M}^i||_2 + ||\boldsymbol{M}^{i+1}||_2)}$ and $\beta^{i+1} = \frac{\boldsymbol{M}^i}{(||\boldsymbol{M}^i||_2 + ||\boldsymbol{M}^{i+1}||_2)}$). Then, if we let the correction between successive steps be equal to $\beta^i \Delta\hat{\mathbf{x}}_{t+1}^i$, one can ensure that (12) is satisfied. Verification proof can be found in Appendix B.2. Pseudo code to enforce inextensibility while preserving momentum can be found in Appendix B.3.

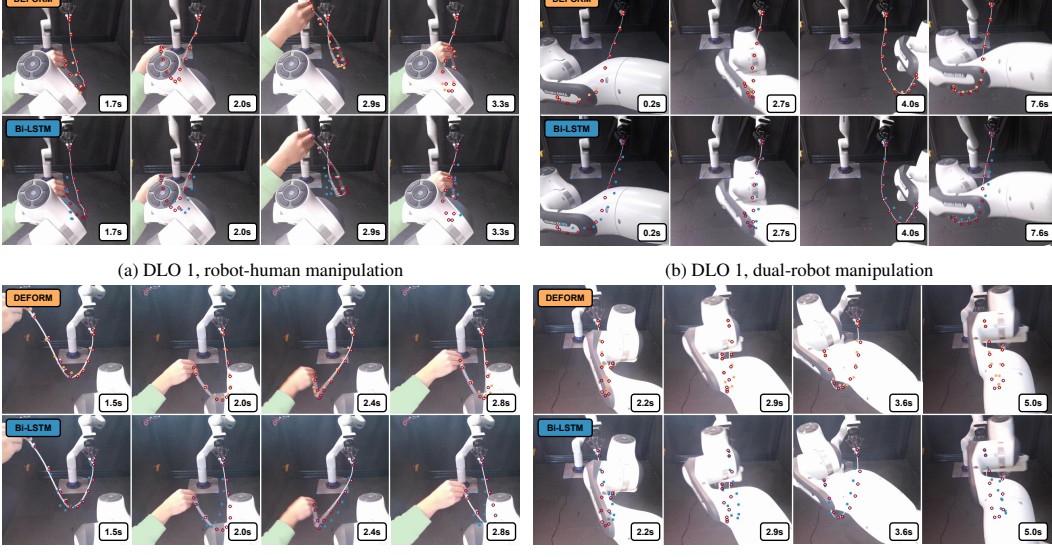

(a) DLO 1, robot-human manipulation

(b) DLO 1, dual-robot manipulation

(c) DLO 2, robot-human manipulation

(d) DLO 2, dual-robot manipulation

Figure 3: Visualization of the performance of DEFORM and Bi-LSTM in predicting trajectories for DLOs 1 and 2. The ground truth vertices of the DLOs are marked with red hollow circles. The predicted vertices are marked with orange solid circles for DEFORM and blue solid circles for the Bi-LSTM model, respectively.

# 5 Experiments and Results

This section covers the real-world evaluation of DEFORM, including: 1) a quantitative comparison of modeling accuracy with state-of-the-art methods, 2) computational speed assessment, 3) an ablation study on parameter auto-tuning, residual learning, and multi-step training, 4) using DEFORM to track DLOs with an RGB-D camera, and 5) a shape matching manipulation demonstration. Code and data will be provided upon final paper acceptance. Additional information is in Appendix C.

## 5.1 Experiment Setup:

**Hardware Setup.** Three distinct cables and two distinct ropes were constructed to validate the modeling accuracy of DEFORM. Spherical markers were attached to the DLOs and are used to create training and evaluation datasets. We set the number of vertices equal to the number of MoCap markers. We use an OptiTrack motion capture (MoCap) system to obtain the ground truth vertex locations. Because the markers are incorporated during training and evaluation, their effect on the dynamics is incorporated into all subsequent evaluations. Note that for real-world applications, the markers would not be needed. A Franka Emika Research 3 robot and a Kinova Gen3 robot are used for dual manipulation.

**Dataset Collection and Training.** For each DLO, we collect 350 seconds of dynamic trajectory data in the real-world using the motion capture system at a frequency of 100 Hz. The model for each of the five DLOs is trained separately using the trajectory data collected from our real-world experiments. The dataset is split with 80% for training and 20% for evaluation. In training, models are trained using the same 100 steps (1s) of dynamic motion, which included slow and fast DLO motions. The L1 loss over the 100 step prediction horizon is used for training. For evaluation, the trained model is used to predict DLO behavior for 500 steps without accessing ground truth.

**Baseline Comparisons. Physics-Based Modeling:** The XPBD [21] approach is constructed using PBD, formulating constraints quasi-statically to model stretching and bending behavior. The DRM [19] method is rooted in the concept of directional diminished rigidity such that the dynamics of a vertex is dependent on its configuration relative to the gripper's configuration. The original DER [13] modeling approach is included as a baseline to demonstrate its performance when compared to other analytical methods and to DEFORM. **Learning-Based Modeling:** Yan et al. [6] and

Wang et al. [15] use Bi-LSTM and GNN to represent a DLO and reason about interactions between the vertices respectively. **Physics-based models with enhancements:** To further demonstrate the capabilities of DEFORM, we enhanced XPBD and DRM with residual learning and material property parameter tuning, which we denote as XPBD-NN and DRM-NN, respectively.

## 5.2 Modeling Results

Table 1 (**Left**) summarizes the results of using DEFORM when compared to baselines using the average L1 loss. Performance is evaluated when predicting the state of the wire over a 5s prediction horizon. Note that DEFORM outperforms all baselines for each of the five evaluated cables. An illustration of the performance of these methods can be found in Figure 3. Next, we compare computational speed for one step inference. Table 1 (**Right**) indicates that Bi-LSTM has the fastest speed. Note that DEFORM is able to generate single step predictions at a 100Hz frequency.

Table 1: **Left:** Baseline comparison results for modeling real-world DLO. **Right:** Computational Speed.

| Method | Modeling Accuracy ($10^{-2}$m) | | | | | Computational Speed ($10^{-2}$s) | | | | |
|---|---|---|---|---|---|---|---|---|---|---|
| | 1 | 2 | 3 | 4 | 5 | 1 | 2 | 3 | 4 | 5 |
| XPBD[21] | 4.00 | 3.85 | 3.80 | 3.62 | 4.35 | 1.45 | 1.38 | 1.33 | 1.31 | 1.36 |
| DRM[19] | 3.64 | 4.16 | 3.21 | 4.02 | 4.19 | 0.45 | 0.44 | 0.43 | 0.44 | 0.41 |
| Bi-LSTM[6] | 1.98 | 1.75 | 1.10 | 1.75 | 1.88 | **0.04** | **0.03** | **0.03** | **0.03** | **0.03** |
| GNN[15] | 3.41 | 2.23 | 2.15 | 2.49 | 2.68 | 1.69 | 1.67 | 1.59 | 1.60 | 1.55 |
| DER[13] | 1.96 | 1.91 | 1.86 | 1.50 | 1.65 | 1.81 | 1.77 | 1.76 | 1.79 | 1.73 |
| NN-XPBD | 2.50 | 2.39 | 2.18 | 2.62 | 2.73 | 1.99 | 1.85 | 1.91 | 1.86 | 1.83 |
| NN-DRM | 2.91 | 3.22 | 2.64 | 2.99 | 3.31 | 0.92 | 0.82 | 0.81 | 0.84 | 0.79 |
| DEFORM | **1.01** | **0.97** | **0.77** | **0.85** | **0.99** | 0.97 | 0.92 | 0.94 | 0.95 | 0.92 |

**Ablation Study.** Results are summarized in Table 2 and show the effect of each DEFORM contribution. Note that a standard DER method is included as a baseline and does not include any of the DEFORM contributions. The results of the study show that residual learning has the biggest improvement on the baseline performance, but the auto-tuning of material properties has a non-trivial affect as well. Further,

Table 2: Ablation Study.

| Method | Accuracy ($10^{-2}$m) | | |
|---|---|---|---|
| | 1 | 2 | 3 |
| DER | 1.96 | 1.91 | 1.86 |
| W/O Residual Learning | 1.54 | 1.77 | 1.42 |
| W/O System ID | 1.21 | 1.32 | 1.09 |
| Original Inextensibility | 1.65 | 1.23 | 1.00 |
| DEFORM | **1.01** | **0.97** | **0.77** |

the improved inextensibility enforcement makes a large improvement for more compliant DLOs, such as DLO1. This is because the more compliant DLOs swing more during dynamic behavior, so ensuring momentum is conserved greatly improves prediction accuracy. The results for DLO4 and DLO5, as well as an additional ablation study, are in Appendix C.3 and C.4, respectively. Note that the stiffnesses of the tested DLOs can be found in Appendix Table 4.

**State Estimation.** We next evaluate the performance of DEFORM when it is used in concert with sensor observations for state estimation. These experiments included occlusions of the DLO that arose from the manipulators or humans moving in front of the sensor. In this experiment, we compare how well the state estimation framework described in Appendix D does when using the DEFORM model versus the Bi-LSTM model. Bi-LSTM was chosen because it was the second best model in terms of model accuracy. We also compare how well each of these methods do with varying frequencies of sensor measurement updates. As in the earlier experiment, we evaluate the L1 norm of the prediction over a 5s long prediction horizon. Experimental results are shown in Figure 4. Unsurprisingly, each method improves when

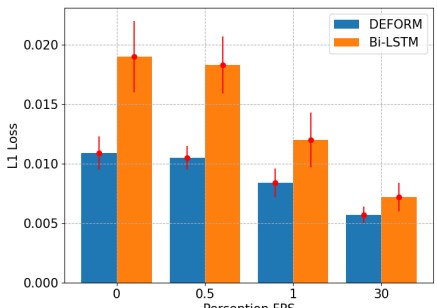

Figure 4: The average L1 loss (m) while performing state estimation.

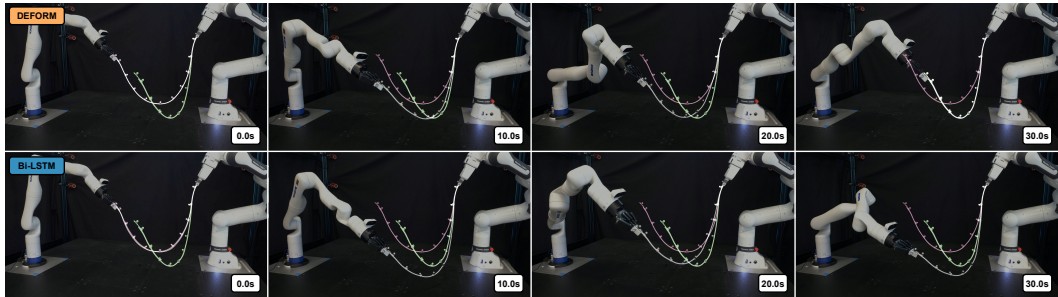

Figure 5: A time lapse comparison of the DEFORM model and the Bi-LSTM model when each is used to perform a shape matching task. The goal configuration is green whereas the start configuration is pink. The DEFORM model enables the planning algorithm to successfully complete the task whereas the planning algorithm using the Bi-LSTM model fails to complete the task.

given access to more frequent sensor updates. This illustrates that the state estimation algorithm described in Appendix D.2 behaves appropriately. Notably, the DEFORM model with no state estimation does better than the Bi-LSTM model even when the Bi-LSTM model is given access to 1 fps updates. The method using the DEFORM model with access to 30 fps sensor measurements is the most effective.

**3-D Shape Matching Manipulation.** To demonstrate the practical applicability of DEFORM, we illustrate its use for DLO 3-D shape matching tasks in both simulation and the real world. This experiment uses ARMOUR [37], a receding-horizon trajectory planner and tracking controller framework, to manipulate DLOs. ARMOUR uses either DEFORM or Bi-LSTM to predict the manipulated DLO state and tries to match the final DLO configuration with a desired configuration. Details of combining ARMOUR with the DLO simulators are in Appendix C.6. We compare the performance of each method on 100 random cases in a PyBullet simulation and 20 random cases in the real world. If the Euclidean distance between each individual pair of ground truth and desired DLO vertices is less than 0.05 meters, then the trial is a success. ARMOUR is given 30 seconds to complete the task, otherwise the trial is marked as a failure. The results are summarized in Table 3 and Figure 5 shows one of the real-world results. A video of these experiments can be found in the supplementary material.

Table 3: Shape matching manipulation success rate

| Method | Success Rate | |
| --- | --- | --- |
| | Real | Sim |
| Bi-LSTM | 10/20 | 78/100 |
| DEFORM | **17/20** | **90/100** |

## 6 Conclusion and Limitations

This paper presents DEFORM, a method that embeds residual learning in a novel differentiable Deformable Linear Object (DLO) simulator. This enables accurate modeling and real-time prediction of dynamic DLO behavior over long time horizons during dynamic manipulation. To demonstrate DEFORM's efficacy, this paper performs a comprehensive set of experiments that evaluate accuracy, computational speed, and its usefulness for manipulation and perception tasks. When compared to the state-of-the-art, DEFORM is more accurate while being sample efficient without sacrificing computational speed. This paper also illustrates the practicality of applying DEFORM for state estimation to track an occluded DLO during dynamic manipulation. Furthermore, DEFORM is combined with a manipulation planning and control framework for 3D shape matching of a DLO in both simulations and real-world scenarios, achieving a higher task success rate compared to the closest-performing baseline.

**Limitations.** We recognize several opportunities to enhance DEFORM. Currently, DEFORM could be improved by adding contact modeling to accurately predict interactions between different components under load [38]. Additionally, DEFORM does not model multi-branched DLOs, which are common in wire-harness assembly [39]. Furthermore, DEFORM assumes that the ends of the DLO are grasped, which could restrict manipulation dexterity. We look forward to addressing these aspects in our future work to improve DEFORM model's accuracy and applicability.

**Acknowledgments**

The authors would like to gratefully thank reviewers for giving useful comments. This work is supported by the Ford Motor Company and the Air Force Office of Scientific Research under the Award No: MURI FA9550-23-1-0400.

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

# A    Additional DER Background

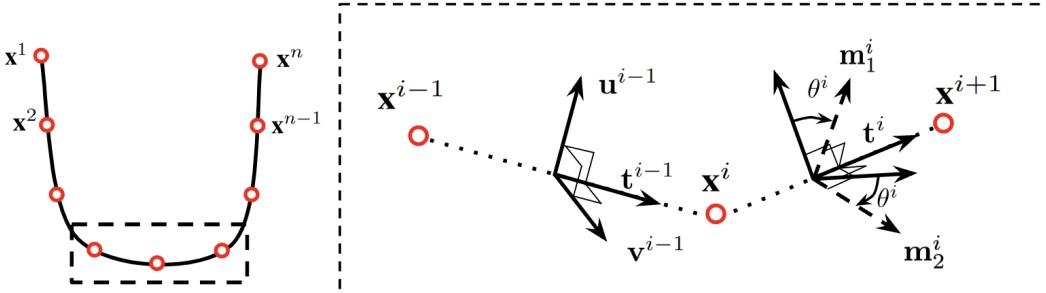

Figure 6: An illustration of DER coordinate frames.

**Bishop Frame.** To accurately approximate the potential energy of the DLO that arises due to bending or twisting, DER first assigns Bishop frames $\{\mathbf{t}^i, \mathbf{u}^i, \mathbf{v}^i\}$ as reference frame onto edge $i$. $\mathbf{t}^i \in \mathbb{R}^3$ represents the unit tangent vector along edge $i$. The vector $\mathbf{u}^i \in \mathbb{R}^3$ is defined iteratively through a rotation operation $\mathbf{u}^i = \mathbf{R}^i \cdot \mathbf{u}^{i-1}$. This rotation is governed by a rotation matrix $\mathbf{R}^i \in \mathbb{R}^{3 \times 3}$, which satisfies following conditions: $\mathbf{t}^i = \mathbf{R}^i \cdot \mathbf{t}^{i-1}$ and $\mathbf{t}^{i-1} \times \mathbf{t}^i = \mathbf{R}^i \cdot (\mathbf{t}^{i-1} \times \mathbf{t}^i)$. $\mathbf{v}^i$, orthogonal to $\mathbf{t}^i$ and $\mathbf{u}^i$, is defined by $\mathbf{v}^i = \mathbf{t}^i \times \mathbf{u}^i$. This transportation results in the least amount of twist, making it ideal for adapting the Bishop frame as the most natural reference frame. Further details of the Bishop frame can be found in [40].

**Material Frame.** DER further introduces a scalar $\theta^i$ to measure the rotation of material frames $\{\mathbf{t}^i, \mathbf{m}_1^i, \mathbf{m}_2^i\}$ relative to the Bishop frame along $\mathbf{t}^i$. $\mathbf{m}_1^i, \mathbf{m}_2^i$ are defined as following

$$\mathbf{m}_1^i = \cos \theta^i \cdot \mathbf{u}^i + \sin \theta^i \cdot \mathbf{v}^i, \quad \mathbf{m}_2^i = -\sin \theta^i \cdot \mathbf{u}^i + \cos \theta^i \cdot \mathbf{v}^i. \tag{13}$$

The material frames are used in (1) to obtain potential energy. An illustration of Bishop frame and material frame can be found in Figure 6. More relevant details can be found in [13, Section 4].

# B    Algorithmic Details

## B.1    DNN Structure

We built our DNN using a Graph Neural Network (GNN) [5]. This is motivated in part by recent work that has illustrated how to successfully apply GNNs to model the spatial relationship between vertices in a DLO [15]. We adapt the default implementation of a graph convolution network (GCN) [41] found in the PyTorch Geomeric library [42]. GCNs have demonstrated their ability to reduce computational costs while effectively extracting informative latent representations from graph-structured data. $\hat{\mathbf{X}}_t$ and $\hat{\mathbf{V}}$ are feature-wise concatenated as $(\hat{\mathbf{X}}_t, \hat{\mathbf{V}}) \in \mathbb{R}^{n \times 6}$ and are the input of the GCN. We set the feature dimension to 32 for message passing, allowing each node to receive information from its local neighborhood. We aggregate each node's neighbors' features using summation. The outputs of the GCN are flattened and decoded by a MLP constructed with two linear layers with a Rectified Linear Unit (ReLU) in the middle.

## B.2    Verification of Improved Inextensiblity Enforcement with PBD

First, note that (9) applies only to successive vertices. This simplifies the conservation of momentum equation(12) so that it only involves successive vertices, as shown below:

$$\boldsymbol{M}^i \cdot \Delta \hat{\mathbf{x}}_{t+1}^i + \boldsymbol{M}^{i+1} \cdot \Delta \hat{\mathbf{x}}_{t+1}^{i+1} = \mathbf{0} \tag{14}$$

$$\mathbf{r}^i \times (\boldsymbol{M}^i \cdot \Delta \hat{\mathbf{x}}_{t+1}^i) + \mathbf{r}^{i+1} \times (\boldsymbol{M}^{i+1} \cdot \Delta \hat{\mathbf{x}}_{t+1}^{i+1}) = \mathbf{0} \tag{15}$$

The complete proposed solution, as detailed in Section 4.3, can be expressed as:

$$\Delta \hat{\mathbf{x}}_{t+1}^i = \frac{\boldsymbol{M}^{i+1}}{(||\boldsymbol{M}^i||_2 + ||\boldsymbol{M}^{i+1}||_2)} \cdot C(\hat{\mathbf{x}}_{t+1}^i, \hat{\mathbf{x}}_{t+1}^{i+1}) \frac{\hat{\mathbf{x}}_{t+1}^{i+1} - \hat{\mathbf{x}}_{t+1}^i}{||\hat{\mathbf{x}}_{t+1}^{i+1} - \hat{\mathbf{x}}_{t+1}^i||_2} \tag{16}$$

$$\Delta\hat{\mathbf{x}}_{t+1}^i = \frac{\boldsymbol{M}^i}{(||\boldsymbol{M}^i||_2 + ||\boldsymbol{M}^{i+1}||_2)} \cdot C(\hat{\mathbf{x}}_{t+1}^i, \hat{\mathbf{x}}_{t+1}^{i+1}) \frac{-(\hat{\mathbf{x}}_{t+1}^{i+1} - \hat{\mathbf{x}}_{t+1}^i)}{||\hat{\mathbf{x}}_{t+1}^{i+1} - \hat{\mathbf{x}}_{t+1}^i||_2} \tag{17}$$

Expanding (14) with (16) and (17):

$$\boldsymbol{M}^i \cdot \Delta\hat{\mathbf{x}}_{t+1}^i + \boldsymbol{M}^{i+1} \cdot \Delta\hat{\mathbf{x}}_{t+1}^{i+1} = \frac{\boldsymbol{M}^i \cdot \boldsymbol{M}^{i+1}}{(||\boldsymbol{M}^i||_2 + ||\boldsymbol{M}^{i+1}||_2)} \cdot C(\hat{\mathbf{x}}_{t+1}^i, \hat{\mathbf{x}}_{t+1}^{i+1}) \frac{\hat{\mathbf{x}}_{t+1}^{i+1} - \hat{\mathbf{x}}_{t+1}^i}{||\hat{\mathbf{x}}_{t+1}^{i+1} - \hat{\mathbf{x}}_{t+1}^i||_2}$$
$$- \frac{\boldsymbol{M}^{i+1} \cdot \boldsymbol{M}^i}{(||\boldsymbol{M}^i||_2 + ||\boldsymbol{M}^{i+1}||_2)} \cdot C(\hat{\mathbf{x}}_{t+1}^i, \hat{\mathbf{x}}_{t+1}^{i+1}) \frac{\hat{\mathbf{x}}_{t+1}^{i+1} - \hat{\mathbf{x}}_{t+1}^i}{||\hat{\mathbf{x}}_{t+1}^{i+1} - \hat{\mathbf{x}}_{t+1}^i||_2} \tag{18}$$

Since DLOs are linear and symmetric about their center axis, their mass matrices are also symmetric [43]. Consequently, $\boldsymbol{M}^{i+1} \cdot \boldsymbol{M}^i = \boldsymbol{M}^i \cdot \boldsymbol{M}^{i+1}$, allowing us to cancel both terms in (18) , resulting in zero linear momentum change. A key observation here is that (16) and (17) are collinear along the edge $i$. Therefore, $\Delta\hat{\mathbf{x}}_{t+1}^i$ and $\Delta\hat{\mathbf{x}}_{t+1}^{i+1}$ will not change orientation of edge $i$, naturally satisfies (15).

### B.3 Summary and Discussion of Improved Inextensiblity Enforcement with PBD

Algorithm 2 summarizes our proposed method to enforce inextensibility while preserving momentum. Note that in practice, the while loop in Algorithm 2 typically converges after two iterations for $\epsilon = 0.05$. Once we have the output from Algorithm 2, we update the velocities of the vertices to reflect the new vertex locations (i.e., Line 5 of Algorithm 1). Further, as shown in Table 6, skipping Algorithm 2 in Algorithm 1 and relying solely on DNNs to capture the in-

---

**Algorithm 2** Enforcing Inextensibility with Momentum Preserving PBD

---

**Require:** $\hat{\mathbf{X}}_{t+1}$ and $\epsilon > 0$
 1: **while** any $C(\hat{\mathbf{x}}_{t+1}^i, \hat{\mathbf{x}}_{t+1}^{i+1}) > \epsilon$ **do**
 2:    **for** $i = 2$ **to** $n - 1$ **do**
 3:      $\hat{\mathbf{x}}_{t+1}^i = \hat{\mathbf{x}}_{t+1}^i + \beta^i \Delta\hat{\mathbf{x}}_{t+1}^i$    ▷ (11)
 4:    **end for**
 5: **end while**
 6: **return** : $\hat{\mathbf{X}}_{t+1}$    ▷ Updated Vertices

---

extensibility of DLOs can lead to simulation instability. This instability arises because modeling stiff behavior, such as the inextensibility of DLOs, makes the learning process sensitive to inputs and hinders effective gradient propagation. A key issue with using neural network predictions in position-based dynamics (PBD) is that the dynamics/data lie on a low-dimensional manifold within a high-dimensional space. Over several open-loop predictions, the network's prediction errors can push the system state away from this manifold, leading to deterioration in the network's predictions. This occurs because the network has never encountered these "off-manifold" points during training. Algorithm 2 is designed to ensure that model inputs remain on the manifold on which the training data was collected, thereby mitigating these issues.

### B.4 Training Setup

Let $\mathbf{U}_{1:T-1}$ denote the set of inputs applied between times $t = 1$ and $t = T - 1$, and let $\mathbf{X}_{1:T} = \{\mathbf{X}_1, \mathbf{X}_2, \ldots, \mathbf{X}_T\} \in \mathbb{R}^{T \times n \times 3}$ denote the associated ground truth trajectory of the DLO from times $t = 1$ to $t = T$. With known $\mathbf{X}_1$, $\mathbf{V}_1$ and $\mathbf{U}_{1:T-1}$, Algorithm 1 can be applied recursively to generate predicted associated trajectory $\hat{\mathbf{X}}_{2:T}$. Let $\phi$ denote parameters of DNN. The objective of training is to solve the following optimization problem:

$$\min_{\boldsymbol{\alpha},\boldsymbol{\phi}} \quad \sum_{t=1}^{T-1} ||\mathbf{X}_{t+1} - \hat{\mathbf{X}}_{t+1}||_2 \tag{19}$$

By taking advantage of DEFORM's differentiability, T can be set to values greater than 2 to capture long-term behavior. This multi-step training pipeline results in higher prediction accuracy than the single-step training pipeline, as shown in Table 6.

### B.5 Discussion of Neural Network Utilization in (7)

This section provides an additional discussion of using neural networks in (7) and (8). Incorporating a GNN enhances prediction accuracy by addressing inaccuracies arising from spatial and

temporal discretization. This network can effectively identify and adjust for these imperfections, as demonstrated in the ablation study. As the training involves a multi-step process, the GNN allows DEFORM to leverage gradients backpropaged from future prediction losses, which enables better informed and proactive corrections. GNN can implicitly capture variations (e.g., shearing) in residual energy that are not directly modeled by the traditional DER framework.

However, the integration of a neural network also introduces several challenges. Primarily, it increases the computational load. Moreover, the addition of a neural network, particularly in capturing and correcting impulse vectors, introduces complexity in ensuring that the network's outputs remain physically plausible and consistent with the DER's modeling output.

### B.6 Comparison of DER to other DLO dynamics formulations

Originally derived from computer graphics, the DER approaches uses Lagrangian mechanics to model the dynamics of a chain of point masses, employing a unique choice of coordinates that effectively captures bending and twisting dynamics. This method contrasts with recent studies that apply Newtonian multi-body dynamics using Featherstone's algorithms [44] and [45]. For instance, [45] models DLO dynamics using a chain of rigid bodies defined in minimal coordinates like angles between body elements, which simplifies the computation by making forward dynamics explicit and eliminating the need for implicit optimization or additional constraint enforcement algorithms. While both Newtonian and Lagrangian dynamics conserve energy, their methods for deriving conservative forces differ, necessitating a deeper discussion on the advantages and limitations of each approach.

From a Lagrangian perspective, DER's innovative coordinate choice and post-integration constraint enforcement allow for the modeling of complex, stiff behaviors through various optimization solvers, enhancing stability and accuracy crucial for DLO simulations. The accurate and stable model offers a high-fidelity prior for learning residual behavior, significantly enhancing sampling efficiency and performance as shown in the ablation study. However, this can introduce significant computational complexity and burden due to the need to solve implicit optimization problems and enforce constraints. In contrast, Newtonian dynamics offer simpler, faster computations due to their explicit nature and minimal coordinate system, facilitating the integration of complex neural networks. However, this can sometimes result in simulation instabilities, particularly when modeling stiff materials, and may struggle to capture complex deformable behaviors without extensive modifications. The pros and cons of each method illustrate a clear trade-off between computational simplicity and the ability to handle complex behaviors. Overall, there are certainly benefits to using Newtonian multi-body dynamics that warrant further exploration.

## C Experimental Details

### C.1 Hardware Parameters

We use an OptiTrack motion capture (Mocap) system to obtain the ground truth vertex locations for DLOs as depicted in Figure 7. Spherical markers with a diameter of 7.9 mm and weight of 0.4 g, are attached to the DLOs using the OptiTrack tracking system. Ten Flex3 cameras capture the motion of the markers at a frequency of 100Hz, with a positional error of less than 0.3mm. For perception with an RGB-D camera, we use an Azure Kinect DK with a resolution of 720 x 1080 and a frequency of 30 Hz. Three distinct cables and two distinct ropes were constructed as shown in Figure 7. The physical properties of each wire are outlined in Table 4. These properties include the length, weight, and stiffness of each DLO, as well as the number of Mocap markers attached to it. The stiffness of each DLO is ranked on a relative scale.

### C.2 Software Implementation:

All experiments were conducted in Python, on an Ubuntu 20.04 machine equipped with an AMD Ryzen PRO 5995WX CPU, 256GB RAM, and 128 cores. All training pipelines were built within

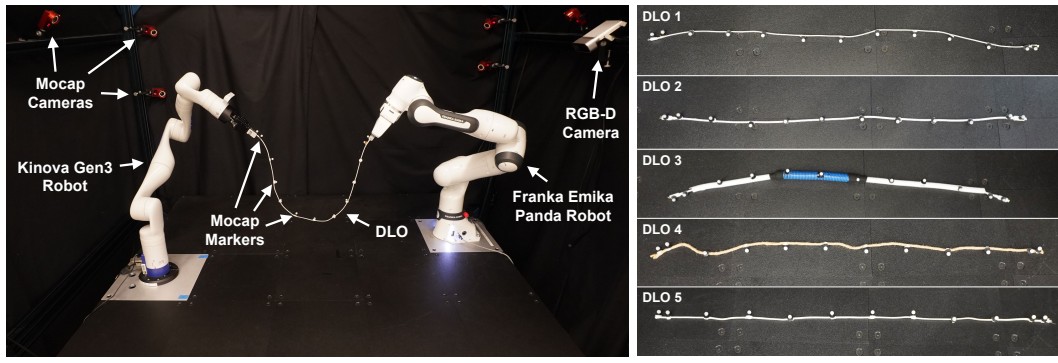

Figure 7: **Left:** An illustration of the experimental setup. **Right:** An illustration of the DLOs that are used to evaluate and compare the performance of DEFORM with various state-of-the-art DLO modeling methods.

Table 4: Material Properties

| Name | Length [m] | Weight [g] | Stiffness | # Mocap Markers |
|---|---|---|---|---|
| DLO 1 | 1.152 | 34.5 | 3 | 13 |
| DLO 2 | 0.996 | 65.2 | 4 | 12 |
| DLO 3 | 0.998 | 96.8 | 5 | 12 |
| DLO 4 | 0.973 | 22.0 | 1 | 12 |
| DLO 5 | 0.988 | 19.2 | 2 | 12 |

the PyTorch training framework. We implement DEFORM with PyTorch and use the Levenberg-Marquardt algorithm as the solver for Theseus. We utilize PyTorch for training and Numpy for non-batched prediction. We initialize the length and mass parameters of the DLO according to Table 4. The other material properties are initialized randomly and learned during the training process. We use SGD optimizer with $10^{-4}$ learning rate for training.

## C.3    Ablation Study: DLO4 and DLO5

As a supplement for Table 2, the ablation study of DLO4 and DLO5 is shown in Table 5.

## C.4    Ablation Study: Inextensibility Enforcement with Learning and Single-step Training

We conducted a further ablation study on training DEFORM without the improved inextensibility enforcement and training DEFORM using only a single-step prediction as shown in Table 6. When DEFORM is trained without enforcing inextensibility, simulation instability results in very high prediction loss, highlighting the importance of properly enforcing inextensibility over long time horizons. Additionally, training DEFORM using only a single-step prediction results in lower long-term prediction accuracy compared to training DEFORM using a 100-step prediction, demonstrating the importance of DEFORM's differentiability for accurate long-term predictions.

Table 5: Ablation Study with DLO 4 and DLO 5.

| Method | Accuracy ($10^{-2}$m) 4 | 5 |
|---|---|---|
| DER | 1.50 | 1.65 |
| W/O Residual Learning | 1.33 | 1.26 |
| W/O System ID | 1.02 | 1.10 |
| Original Inextensibility | 1.29 | 1.58 |
| DEFORM | **0.850** | **0.987** |

Table 6: Additional Ablation Study.

| Method | Accuracy ($10^{-2}$m) | | |
| --- | --- | --- | --- |
| | 1 | 2 | 3 |
| W/O Enforcing Inextensibility with PBD | $7.7 \times 10^5$ | $260 \times 10^5$ | $310 \times 10^5$ |
| Single Step Prediction Training | 1.82 | 1.74 | 1.79 |
| DEFORM | **1.01** | **0.97** | **0.77** |

## C.5 Effect of Number of Vertices: Accuracy and Computational Efficiency

To further investigate the impact of vertices number on accuracy and computational efficiency, we conducted additional experiments using DLO 1. Figure 8 illustrates that increasing the number of vertices enhances prediction accuracy by allowing DEFORM to capture finer details due to improved resolution. However, this increase in detail is at the cost of reduced computational efficiency. In our case we chose the number of vertices to achieve optimal prediction accuracy while ensuring real-time inference capabilities at a 100Hz frequency.

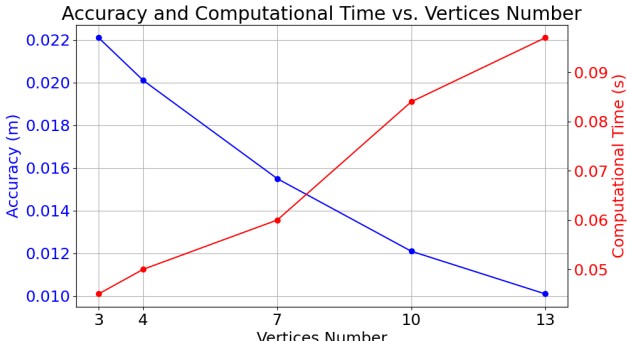

Figure 8: Accuracy and Compuational Time vs. Vertices Number

## C.6 ARMOUR

To perform a shape matching task, we rely upon ARMOUR[37], an optimization-based motion planning and control framework. The goal of the shape matching task is to use a robot arm to manipulate the DLO from an initial configuration to a predefined target configuration. To accomplish this goal, ARMOUR performs planning in a receding horizon fashion. During each planning iteration, ARMOUR selects a trajectory to follow by solving an optimization problem. ARMOUR represents joint trajectories of the robot as polynomials, where the coefficients are considered as the decision variables of the optimization. One end of the rope is rigidly attached to the end effector of the robot, whose configuration can be computed by forward kinematics. By plugging the end effector configuration at the end of the trajectories into DEFORM, we are able to get a prediction of the rope configuration at the end of the robot motion. The cost function for the optimization problem to minimize is then defined as the norm distance between the predicted rope configuration and the target rope configuration. Joint position, velocity, and torque limits of the arm are considered as constraints of the optimization problem in ARMOUR's formulation, which ensures that the robot motion is feasible. The solution to this optimization problem is then the joint trajectories of the robot that manipulate the rope to approach the desired configuration. To command the robot to track the solution trajectories, we use the controller proposed in ARMOUR. This process is conducted repeatedly until the desired configuration is reached, or a planning iteration limit is reached resulting in a failure. More details about ARMOUR's trajectory parameterization and associated closed loop controller can be found in [37, Section IX]. The cost function minimizes the distance between the

predicted DLO configuration given a chosen robot trajectory and a target DLO configuration. The predicted state of the DLO is computed via a DLO modeling technique.

# D  DLO Tracking with Modeling

This section first discusses the difficulties of incorporating existing framework with modeling for long-time DLO tracking under occlusion. It then proposes a novel perception pipeline, which is utilized in Section 5.2.

## D.1  DLO Tracking Review

If a DLO is fully observable, then current state-of-the-art methods estimate the location of the DLO's vertices by applying a Gaussian Mixture Model (GMM), performing clustering, and then using Expectation-Maximization [46, 47, 48, 49, 50, 51, 52, 53]. The output of this GMM algorithm is the mean locations in $\mathbb{R}^3$ of each of the Gaussians, which is then set equal to the vertex locations of the DLO. However, in practical applications, occlusion of DLOs during manipulation often leads to perception challenges, which complicates accurate prediction. In particular, due to occlusions, one cannot simply set the number of mixtures in the GMM equal to the number of vertices of the DLO. Doing so results in the vertices being incorrectly distributed only to the unoccluded parts of the DLO. Some of the aforementioned methods[49, 50] have been applied to perform tracking of DLOs under occlusion. Typically, this is done by leveraging short time horizon prediction with geometric regularization using prior observations. [49, 50, 51, 52, 53] Though powerful, to work accurately, these methods require frequent measurement updates and can struggle in the presence of occlusions for long time horizons. Recent research has explored particle filtering within a lower-dimensional latent space embedding and applied learning-based techniques for shape estimation under occlusion [54, 55]. Each of these methods rely upon different models for DLOs that tend to have numerical instabilities when used for prediction. As a result, these perception methods require high frequency sensor measurement updates to behave accurately. This paper illustrates that our proposed model allows us to adapt DEFORM's long time horizon prediction capability to relax the frequency of sensor updates, which reduces the overall computational cost of tracking DLOs in the presence of occlusions.

## D.2  DLO Tracking with DEFORM

This section describes how we can use DEFORM to perform robust DLO tracking. Notably, DE-FORM enables us to deal with occlusions without requiring a frame-by-frame sensor update of the state of the DLO. In particular, this is possible due to our model accurately predicting the state of the DLO over long time horizons. As a result, we utilize a GMM model because it is independent of time. The state estimation approach is outlined in Algorithm 3. Additionally, Algorithm 4 summarizes the perception pipeline that describes the tracking of DLOs with state estimation and initial state estimation using DEFORM.

## D.3  State Estimation in the Presence of Occlusions

Suppose that we are given access to an RGB-D sensor observing our DLO during manipulation and suppose that we translate these measurements at time step $t$ into a point cloud which we denote by $\mathbf{S}_t = \{s_t^1, s_t^m, \ldots, s_t^M\} \in \mathbb{R}^{\mathrm{M}\times 3}$.

To address the limitations of the algorithms discussed in Section 2, we leverage our predictions generated by applying Algorithm 1 as follows. We first filter the point cloud at time $t$ using our predicted data. If any element of the point cloud is beyond some distance from our prediction, $\hat{\mathbf{X}}_t$, then we remove it. Let $\tilde{\mathbf{S}}_t$ denote the remaining points in the point cloud. Next, we detect which of the vertices of the DLO are unoccluded from the RGB-D sensor by checking if their predicted depth is close to their observed depth in the sensor. Suppose the number of vertices that are unoccluded is $\tilde{n}$.

---

**Algorithm 3** State Estimation with Presence of Occlusion

---

**Inputs**: $\hat{\mathbf{X}}_{t+1}$                                          ▷ Algorithm. 1

  1: $\mathbf{S}_{T+1} \leftarrow$ RGB-D Camera
  2: $\tilde{\mathbf{S}}_{T+1} \leftarrow$ filter$(\hat{\mathbf{X}}_{t+1}, \mathbf{S}_{T+1})$
  3: Unoccluded $\hat{\mathbf{X}}_{t+1} \leftarrow$ Depth Matching$(\tilde{\mathbf{S}}_{T+1}, \hat{\mathbf{X}}_{t+1})$
  4: $n+1$ groups $\leftarrow$ DBSCAN$(\tilde{\mathbf{S}}_{T+1})$
  5: **for** each group **do**
  6:     GMM Number j $\leftarrow$ Match(Unoccluded $\hat{\mathbf{X}}_{t+1}$, group)
  7:     Mixture Center $\leftarrow$ GMM(group, j)
  8: **end for**
  9: **for** each $\hat{\mathbf{X}}_{t+1}$ **do**
10:     **if** Observed **then**
11:         Vertex $\leftarrow$ Associated Mixture Center
12:     **else**
13:         Vertex $\leftarrow$ Predicted Vertex
14:     **end if**
15: **end for**
16: **return** : Vertices

---

**Algorithm 4** Tracking DLO with Modeling

---

**Initial State Estimation**

**Require:** $\mathbf{S}_1 \leftarrow$ RGB-D Camera
  1: DLO Initial Guess $\leftarrow \mathbf{S}_1$
  2: **while** DLO not Static **do**
  3:     Execute DEFORM                               ▷ Algorithm. 1
  4: **end while**
  5: Return $\hat{\mathbf{X}}_1, \hat{\mathbf{V}}_1 = \mathbf{0}$

**Tracking DLO under Manipulation**

  1: **while** Tracking DLO **do**
  2:     **while** Predicting DLO **do**
  3:         Execute DEFORM                       ▷ Algorithm. 1
  4:     **end while**
  5:     State Estimation and Correction
  6: **end while**

---

Once this is done, we apply DBSCAN to group $\tilde{\mathbf{S}}_t$ into $\tilde{n}+1$ groups. Next, we associate each of the unoccluded vertices with one of the groups by finding the group to which its predicted location is smallest. We then take each group and perform GMM on the group with a number of mixtures equal to the number of vertices j that were associated with that group. Note, each mixture is associated with an unoccluded vertex of the DLO. The new predicted location of the vertex generated by DEFORM is updated by setting the new vertex location equal to the mean of its associated mixture. If a particular vertex was occluded, then its predicted location is left equal to its output from Algorithm 1.

### D.4 Initial State Estimation under Occlusion

Note that in many instances, it can be difficult to estimate the location of the vertices when the sensor turns on and the DLO is occluded. Unfortunately, Algorithm 1 and the state estimation algorithm from the previous subsection require access to a full initial state. Fortunately, we can address this problem by making the following assumption: When the sensor measurements begin, the DLO is static and is only subject to gravity and/or the manipulators which are holding its ends. Under the above assumption, we initialize the DLO using an initial guess that aligns with observed vertices. This initial guess is then forward simulated using Algorithm 1 repeatedly until the DLO reaches a static state. This steady state is then used as the predicted state for all the vertices, including the occluded vertices.

