# OpenReview forum: "Differentiable Discrete Elastic Rods for Real-Time Modeling of Deformable Linear Objects"
_robot-learning.org/CoRL/2024/Conference — CoRL 2024_

### Official Review · Reviewer_ktjq · 2024-07-14
**Nice paper with convincing experiments**

**Originality:** 4
**Technical Quality:** 4
**Clarity Of Presentation:** 4
**Potential Impact:** 3
**Recommendation:** 4
**Confidence:** 4

**Review:**

I enjoyed reading this paper.

**Strength:**
- Proposes a novel method to learn DLO dynamics with DER
- Decently written
- Convincing experiments with various DLOs

**Weaknesses:**
- The introduction of the DER model could be improved, e.g. by adding an illustration of the involved coordinate frames
- **C1: Pros and cons of the choice of MLP inputs/outputs?** In Equation (7), the authors add to the DER model a neural network to learn the error on the impulse vector. The network receives as input and predicts the impulse in Cartesian coordinates. Clearly, this is an obvious way on how to add a network to Equation (7), but I would appreciate if the authors discuss the pros/cons of placing a network at this particular place in the model. After all, the DER model defines the potential function solely in terms of neighboring edge/frame coordinates,

- **C2: Comparison of DER to other DLO dynamics formulations:** Initially emerging from the field of computer graphics, the DER implements Lagrangian dynamics of a chain of point masses with a nifty choice of coordinates. In comparison, the recent work [1] and [2] (see below) learn DLO dynamics by resorting to Newtonian multi-body dynamics following Featherstone's multi-body algorithms. In particular, [2] implements DLO dynamics as a chain of rigid bodies which are described in minimal coordinates (angles between body elements). As the dynamics are described in minimal coordinates, the forward dynamics are explicit such that they do not require solving an implicit optimization problem and one does not need to enforce implicit constraints with an additional algorithm. Also note that both Newtonian and Lagrangian dynamics equations respect energy conservation, they only differ by how conservative impulses/forces are being derived. In this regard, the pros and cons of resorting to Lagrangian mechanics and position-based dynamics requires further discussion.

- **C3: Algorithm for Inextensiblity Enforcement with PBD:** The following statement requires clarification: *"Further, as shown in Table 6, skipping Algorithm 2 in Algorithm 1 and relying solely on DNNs to capture the inextensibility of DLOs can lead to simulation instability. This instability arises because modeling stiff behavior, such as the inextensibility of DLOs, makes the learning process highly sensitive to inputs and hinders effective gradient propagation [8, 11]. "* - A general problem with adding neural network predictions to position-based dynamics (PBD) is that your dynamics/data lie on a low-dimensional manifold inside a high-dimensional space. In turn, over the cause of several open-loop predictions, the network's prediction errors will push the system state away from the manifold on which the data has been collected on. In turn, the network predictions deteriorate as the network has never seen these "off-manifold" points during training. Through Algorithm 2, the authors ensure that model inputs remain on the manifold on which training data has been collected on.

- **C4: Ensuring momentum conservation:** In line 187, the authors propose to weight the state correction term using the inertia matrices of neighboring nodes. I do not see, why this correction term enforces momentum conservation when applying the correction term. Either improve the explanation or plot the change in system momentum before/after the constraint enforcement is being applied with/without using the momentum correction term.


**Minor comments:**
- Line 98: Please emphasize that the velocity $V_t = X_{t} - X_{t-1}/ \Delta t$ is only an approximation of the actual velocity
- Typos, e.g. Appendix B - "Experiemental details"

*[1] -  "Pseudo-rigid body networks: learning interpretable deformable object dynamics from partial observations", Mamedov et al., 2024 - https://arxiv.org/abs/2307.07975*

*[2] - "Learning deformable linear object dynamics from a single trajectory", Mamedov et al., 2024 - https://arxiv.org/abs/2407.03476*

**Quality Of The Limitations Section:**

3

**Questions For Rebuttal:**

**Question - C1:** What are the pros and cons of the particular choice of input/outputs to the residual network? To which extent does the used GNN incorporate the structural knowledge underlying the DER model?

**Question - C2:** With the goal in mind of learning the errors of DLO dynamics with neural networks, what are the pros and cons of formulating dynamics in terms of a potential compared to resorting to Newtonian dynamics algorithms as in [2]?

**Robotics Focus:**

4

**Summary Of Paper:**

The authors propose a discrete-time DLO dynamics model that combines a differntiable discrete elastic rod (DER) dynamics model with neural network regression. The model is derived in three steps:  1. Implement DER dynamics in PyTorch such that it is differentiable wrt. the potential parameter $\theta$ (resorting to implicit differentiation via Theseus) 2. An MLP is added to the DER dynamics model that learns the residual in the impulses and the residual of the numerical integration on the position-level 3. Add constraint enforcing algorithm to ensure inextensibility of the DLO model during integration (similar to [33])

**Summary Of Recommendation:**

The proposed combination of DER dynamics models with neural network regression forms an important contribution towards understanding how to most effectively combine DLO dynamics models with neural networks for planning and control.  The discussion of DER and why the authors chose this particular DLO dynamics model compared to other models needs to be improved.

---

### Official Review · Reviewer_jfrQ · 2024-07-20
**Good paper with extensive validation in experiments**

**Originality:** 4
**Technical Quality:** 4
**Clarity Of Presentation:** 4
**Potential Impact:** 3
**Recommendation:** 3
**Confidence:** 4

**Review:**

The paper presents an advancement in the modelling of deformable linear objects by introducing a hybrid framework that combines the precision of physics-based models with the adaptability of learning-based approaches. The results demonstrate the potential of DEFORM for real-time, accurate, and robust DLO modelling, making it a valuable contribution to the field.

The integration of differentiable Discrete Elastic Rods (DER) with deep learning that leverages the strengths of both physics-based and data-driven models is not new to the field. But the results are good. The framework's ability to provide real-time predictions is significant for practical applications.

My only scepticism about this paper is that it is not a robotics paper, robot are there just to move the DLO around. But of course, this work is beneficial to works that manipulate DLOs. Therefore I would encourage the authors to opensource the code once the paper is accepted

'These objects are often categorized as Deformable Linear Objects (DLOs) because they have complex dynamics due to bending and twisting' No, it is because they are linear and also deformable.

**Quality Of The Limitations Section:**

3

**Questions For Rebuttal:**

How to decide the number of vertices required for one DLO?

Can DEFORM handle twists? It is in the physic model but the demonstrate tasks do not have twist?

**Robotics Focus:**

2

**Summary Of Paper:**

The paper introduces DEFORM, a novel framework combining differentiable physics-based models with learning frameworks to accurately model and predict the dynamic behavior of Deformable Linear Objects (DLOs) such as ropes and cables. The framework promises real-time performance and addresses challenges like numerical integration errors and inextensibility enforcement. The authors present comprehensive experiments to demonstrate DEFORM's superior accuracy, computational speed, and generalizability compared to state-of-the-art methods

**Summary Of Recommendation:**

This is a decent paper for CoRL

---

### Official Review · Reviewer_rJLv · 2024-07-21
**Quick Overview and Initial Impressions of DEFORM Framework for Real-time DLO Modeling**

**Originality:** 4
**Technical Quality:** 3
**Clarity Of Presentation:** 3
**Potential Impact:** 2
**Recommendation:** 3
**Confidence:** 3

**Review:**

### Quality
The quality of the paper is high, with a well-organized structure. The writing is generally clear and technically sound, although some sentences could be broken down for better readability.

### Clarity
The clarity of the paper is commendable. Key concepts and methodologies are explained in a straightforward manner.

### Originality
The originality of the paper is significant. The introduction of DEFORM, a framework that combines differentiable physics-based modeling with residual learning for real-time DLO prediction, represents a novel approach in the field.

### Strengths:
1.  The integration of residual learning into a differentiable simulator for DLOs is a novel concept that addresses existing limitations in real-time modeling and long-term prediction.
2.  The extensive set of experiments validates the framework's efficacy in terms of accuracy, computational speed, and generalizability.

### Weaknesses:
1.  Some sentences are lengthy and complex, which may hinder readability.
2.  Some specific metrics need to be explained.

**Quality Of The Limitations Section:**

3

**Questions For Rebuttal:**

1, Could you clarify the computational efficiency of DEFORM, specifically how it scales with the complexity of the DLOs?

2. Could you provide more details on the manipulation planning and control framework integrated with DEFORM for 3D shape matching?

**Robotics Focus:**

4

**Summary Of Paper:**

This paper introduces DEFORM, a method that integrates residual learning into a novel differentiable simulator designed for Deformable Linear Objects (DLOs). The framework enables precise modeling and real-time prediction of DLO behavior over extended time periods during dynamic manipulation. The efficacy of DEFORM is demonstrated through extensive experiments assessing its accuracy, computational speed, and applicability in manipulation and perception tasks. Compared to existing state-of-the-art methods, DEFORM achieves higher accuracy and sample efficiency, without compromising computational speed. Additionally, DEFORM proves practical for state estimation, particularly in tracking DLOs in occlusion-rich environments. The paper further showcases DEFORM's utility by combining it with a manipulation planning and control framework to achieve 3D shape matching of DLOs in both simulated and real-world settings, resulting in a higher task success rate than the nearest baseline.

**Summary Of Recommendation:**

In summary, the paper on DEFORM presents a novel approach that combines physics-based modeling with learning techniques for real-time modeling of Deformable Linear Objects (DLOs). The experiments conducted demonstrate the effectiveness of DEFORM in terms of accuracy, computational speed, and practical applications in manipulation tasks. While the findings show promising results, it is acknowledged that there are some limitations to be addressed in future work. Therefore, it is recommended that further research be conducted to strengthen the acceptance of DEFORM by addressing these limitations and enhancing its capabilities, particularly in contact modeling and multi-branched DLO modeling. By addressing these areas, DEFORM can be further improved to increase its accuracy and applicability in dynamic DLO modeling.

---

### Author Rebuttal · Authors · 2024-08-13

We thank all reviewers for taking the time to read our paper and provide detailed feedback. We have addressed all weaknesses and rebuttal questions to the best of our ability. Please let us know if your concerns have been addressed and feel free to ask more questions.

The rebuttal file includes the revised manuscript, the previous demo video, and a newly conducted twisting demonstration.

---

### Decision · Program_Chairs · 2024-09-04

**Decision:**

Accept

**Comment:**

This paper addresses the task of modeling Deformable Linear Objects (DLOs), such as ropes and cables, during dynamic motion over long time horizons. It proposes differentiable Discrete Elastic Rods For deformable linear Objects with Real-time Modeling (DEFORM), a framework that combines a differentiable physics-based model with a learning framework to model DLOs accurately and in real-time. Compared to existing state-of-the-art methods, DEFORM achieves higher accuracy and sample efficiency, without compromising computational speed.
Overall, this seems like a good paper. However, issues raised by the reviewers, such as the writing, metrics, implementation details, need to be fixed during the rebuttal, in addition to clarifying certain statements indicated by the reviewers.